# Should my automated car drive as I do? Investigating speed preferences of drivengers in various driving conditions

**Maxime Delmas[1]\*, Valérie Camps[2], Céline Lemercier[1]**

**1** Language and Ergonomics (CLLE) Laboratory, Cognition, Languages, University of Toulouse—Jean Jaurès, Toulouse, France, **2** Toulouse Computer Science Research Institute (IRIT), Paul Sabatier University, Toulouse, France

\* maxime.delmas@univ-tlse2.fr

## Abstract

Studies investigating the question of how automated cars (ACs) should drive converge to show that a personalized automated driving-style, i.e., mimicking the driving-style of the human behind the wheel, has a positive influence on various aspects of his experience (e.g., comfort). However, few studies have investigated the fact that these benefits might vary with respect to driver-related variables, such as trust in ACs, and contextual variables of the driving activity, such as weather conditions. Additionally, the context of intermediate levels of automation, such as SAE level 3, remains largely unexplored. The objective of this study was to investigate these points. In a scenario-based experimental protocol, participants were exposed to written scenarios in which a character is driven by a SAE level 3 AC in different combinations of conditions (i.e., types of roads, weather conditions and traffic congestion levels). For each condition, participants were asked to indicate how fast they would prefer their AC to drive and how fast they would manually drive in the same situation. Through analyses of variance and equivalence tests, results showed a tendency for participants to overall prefer a slightly lower AC speed than their own. However, a linear regression analysis showed that while participants with the lowest levels of trust preferred an AC speed lower than theirs, those with the highest levels preferred an AC speed nearly identical to theirs. Overall, the results of this study suggest that it would be more beneficial to implement a personalization approach for the design of automated driving-styles rather than a one for all approach.

## Introduction

### Background

The technological aspects of automated cars (ACs) are progressing at a rapid pace, and the availability to the general public of vehicles with higher levels of automation draws closer. Their development is expected to reduce the number of accidents, traffic congestion, and even the carbon footprint of this type of transport [1]. Attention is now being partially shifted from

---

**Data Availability Statement:** All data supporting the article are publicly available at https://osf.io/y9s3a/.

**Funding:** This research was supported by a PhD contract funded by the Occitanie Regional Council

---

(France), and the neOCampus project (Paul Sabatier University, Toulouse, France). The funders had no role in study design, data collection and analysis, decision to publish, or preparation of the manuscript.

**Competing interests:** The authors have declared that no competing interests exist.

the sheer technological feasibility to the question of how these ACs should behave in order to optimize their interaction with humans, as this interaction is considered a key issue to ensure that their expected benefits can be realized [2].

As with human drivers, ACs' driving behavior can be defined in terms of "driving-style", characterized by various driving parameters such as driving speed, headway distances or over-taking thresholds [3]. One focus of the scientific literature on the issue of ACs' driving-styles has been to study whether it would be beneficial to the experience of the human behind the wheel (e.g., to his comfort) if the AC mimicked his own driving-style. The expected benefits of such a personalized approach can be explained through the prism of various theoretical models of the driving activity. For instance, the "multiple comfort zones" model [4] specifies that during manual driving, drivers try to maintain 4 major mobility factors within a "comfort zone", differing inter-individually. First, drivers tend to keep space and time safety margins to road edges, obstacles, other vehicles and, in the end, to crash. When adequate safety margins are respected, drivers experience a feeling of control, comfort and safety. Second, drivers tend to have thresholds regarding different aspects of driving that allow keep the ride smooth and comfortable. For instance, while approaching a signalized intersection, drivers tend to pass through if the required deceleration exceeds 3 to 3.5 m/s$^2$ at the time of yellow onset. Third, drivers tend to follow explicit rules (e.g., speed limit) to avoid sanctions, and implicit rules (e.g., speed of other vehicles on the road) to avoid going against social norms. The fourth and last factor corresponds to the fact that travel progresses as expected. Staying in the comfort zone for this mobility factor involves maintaining speed and pace, which means that deceleration can be felt punishing. Through the prism of this model, it can be hypothesized that a personalized automated driving-style, mimicking that of the human behind the wheel, would respect the latter's comfort zones and thereby improve his experience. Conversely, an unfamiliar automated driving-style could violate these comfort zones, for instance by engaging in maneuvers considered as unsafe or uncomfortable. The studies on this subject seem to be in line with these assumptions, as they have shown that in most situations, personalized automated driving-styles are preferred [1,5,6], more accepted [6,7], and are perceived as more trustful [6,8], comfortable [6,8] and safe [6].

Yet, these studies also suggest that the benefits of automated driving-style personalization could vary according to the situations at hand. For instance, participants in one study were driven by a SAE level 2 AC on the right lane of a straight two-lane highway [5]. The results of this study have shown that the preference for the personalized automated driving-style (compared to artificial alternatives) in over-taking situations was more pronounced when there was a vehicle approaching on the left lane at 160 km/h than when no vehicle was travelling on the left lane. Another study has also shown that while young participants preferred a familiar automated driving-style, older participants preferred an unfamiliar one, tending to be faster than their age-affected manual driving-style [1]. Other studies have also shown that the appreciation of automated driving-style could vary according to driving situations such as driving in intersections [9], in adverse weather conditions [10], or in heavy traffic [10]. These variations might be explained by driver-related variables, such as trust in automated driving systems. It has for instance been shown that the participants with the highest level of trust in ACs had their comfort less affected by adverse driving conditions than participants with the lowest level of trust in AC [10]. Calibration of trust in an automated system must be appropriate for the interaction to be safe and effective, since over-trust and mistrust can lead to both the misuse and disuse of the automated system [11,12].

Overall, these results suggest that the benefits of automated driving-style personalization could be modulated with respect to the driving situations encountered, and variables inherent to individuals, such as trust in AC. However, the literature on these modulating effects is still

very scarce, and the influence of most variables remain unexplored. It is crucial to develop our knowledge on these points, since ACs will be used in complex driving environments and by individuals with varying characteristics. Another important gap in the scientific literature on the benefits of automated driving-style personalization concerns the levels of automation studied. In fact, most studies only consider higher levels of driving automation (i.e., SAE levels 4–5 [13]). Therefore, to date, the influence of automated driving-style personalization in the context of intermediate levels of driving automation (i.e., SAE level 1–3) remain largely unknown.

## The present study

The present study was intended to fill some of these gaps. It was conducted online using a scenario-based experimental protocol (see the Method section for more details). Its first main objective was to determine whether the benefits of automated driving-style personalization hold in various driving conditions, and according to trust in ACs. The variables used to characterize driving conditions in this study were road types, weather conditions, and the level of traffic congestion. These variables were selected because they are inherent to the driving activity, and are determinants of its complexity [14]. The second main objective of this study was to investigate the extent to which the benefits of personalizing automated driving-styles generalize to SAE level 3. This level of driving automation allows the driver to delegate control of the vehicle to the automated system, but while remaining able to resume manual driving if necessary. While the automated system is driving, the human behind the wheel is allowed to engage in non-driving-related tasks (e.g., reading) without the need the supervise the automated driving system. The driving activity will thus be shared over time between the vehicle and the human behind the wheel. In this context, the latter will become a "drivenger", alternating between driver and passenger status. The transfer of control of the driving task from the automated system to the human, or "take-over" phase, is highly critical. This can be explained by various phenomenon, such as the loss of situation awareness when engaging in non-driving-related tasks [15]. Additionally, this inherent criticality can be exacerbated by driver-related variables, such as age [16], or environment-related variables, such as weather conditions [16] or traffic congestion [17]. One of the reasons why this level of automation is less studied than levels 4–5 could be that car manufacturers seem reluctant to develop such systems. Indeed, they combine both the potential problems related to the take-over phase, but also the question of the responsibility of the vehicle's behavior, especially in case of a crash. However, since May 17, 2022, SAE Level 3 cars are available in Germany [18]. The deployment of this level of automation is therefore no longer fiction but reality, and it is necessary to develop our understanding of the interaction between humans and such systems. In addition to the multiple comfort zones model [4], the task-capability interface model of the driving process [14] provides insight into why automated driving-style personalization could also have benefits in the context of SAE levels 3, characterized among other things by an alternating control of the driving task between the drivenger and the automated driving system. This model specifies that the difficulty of the driving task is located at the interface between two components. On one hand, it is at the junction with the task demands (i.e., the objective complexity of the driving task), emerging from the combination of different variables: the speed of the vehicle, the presence of other road users, weather conditions, etc. On the other hand, it is at the junction with the driver's capability, which refers to the ability of a driver to apply its competence at a given time. Driver's capability is also influenced by a combination of different variables: age, experience, emotions, perceptual acuity, etc. The difficulty emerging from the transaction between the task demands and the available level of capability can be tackled by drivers from both ends: they can, for example, reduce the speed of the vehicle (i.e., modifying task demand) and/or increase

their vigilance level (i.e., modifying capability). The model also specifies that task difficulty and perceived risk increase as the threshold of inability to meet the demands of the task approaches. Through the prism of this model, the expected benefits of automated driving-style personalization would lie in the fact that a personalized automated driving-style would help match the driver's capabilities to the driving task demands. It could thereby improve his experience by reducing perceived risk or boredom (i.e., respectively due to the task difficulty being too high or too low). It could also improve the take-over performance, transferring back the control of the driving task to the drivenger under conditions corresponding to his capability. Conversely, a non-personalized automated driving-style could lead to an increased risk perception and/or to a dangerous take-over phase.

In summary, the overall objective of this study was to investigate the extent to which, in a SAE level 3 automated car, the willingness for a personalized automated driving-style would hold with respect to various driving conditions and according to trust in ACs. In the context of this study, driving-style was only considered through its speed aspect. It has been shown that the evaluation of driving speed through self-reported questionnaires could be used to replace direct observations [19].

## Method

### Ethics statement

The study was approved by the Research Ethics Committee of the Federal University of Toulouse (ethical identification number 2021–387) and conducted in accordance with the principles expressed in the Declaration of Helsinki and the American Psychological Associations' Ethical Principles of Psychologists and Code of Conduct. All participants gave a written informed consent that they took part in the study voluntarily.

### Participants

The sample consisted of 103 participants (59 females, 42 males, 2 non-specified; $M_{age}$ = 40.66 years, $SD$ = 15.74). Most of them (77.67%) were active drivers, driving more than 1 time a week during the last 6 months. Very few (1.94%) had been involved in a road accident with only material damage during the past year, and none with injured persons or fatalities. Either automatic cruise control or lane centering had already been used by 16.50% of the participants, and automated cruise control coupled with lane centering by 32.04%.

Participants were all French speakers, and were recruited via Facebook groups or by email (i.e., professional and personal networks). The only condition for taking part was to have a valid driver's license. Participants were not remunerated for taking part.

### Materials

**Rationale.** Most studies on driving automation have been based on either real road experiments (e.g., [20]) or in a driving simulator (e.g., [1]). Scenarios-based experimental protocols offer a viable, cost-efficient alternative. This methodology, based on the information integration theory [21,22], allows several variables and their mutual interactions to be investigated at the same time. It relies on written scenarios, where participants are asked to evaluate combinations of variables, rather than individual ones. This methodology has already been used in various research areas [23], including automated driving [10,24].

**Scenario composition.** Twenty-four written scenarios were constructed, according to 3 within-participant variables: type of road (*highway vs. secondary vs. downtown*) × weather conditions (*clear weather vs. very rainy*) × traffic congestion level (*few vehicles vs. many vehicles*).

For each scenario, participants first had to read the text (e.g., "*Charlie is in his partially autonomous vehicle on the highway. The weather is clear. There are few vehicles on the road*"). Then, they had to indicate their anticipated attitudes and behaviors on a visual analogue scale as if they were in the protagonist's shoes. In the present study, an original modification was made to this second phase. Rather than asking participants to indicate their responses on a single item, which is usually done (e.g., [10,24]), they were asked to indicate their responses on two. First, they were asked: "*If you were Charlie, how fast would you prefer the partially autonomous vehicle to drive in this situation*?". Then, they were asked: "*How fast would you drive in the same situation*?". For both questions, they had to indicate their responses on a scale ranging from 0 to 180 km/h, with 1 km/h increments. This change in the experimental protocol allowed for a richer assessment of the anticipated attitudes and behaviors of the participants in each condition, by being able to directly compare the two speeds. The questionnaires were developed on the Qualtrics online platform. The order of presentation of the different scenarios was randomized.

**Scenario instructions.** During the instruction phase, participants were asked to read each scenario of the questionnaire carefully, and to answer by taking into account all the elements contained in the stories. They were informed that they would be able to modify their answers during the familiarization phase of the study, but not during the subsequent phase. The car in which the stories' protagonist was seated was described as partially automated, that is, capable of automatically maintaining the speed and position of the vehicle on the road. Participants were also informed that drivers are allowed to engage in non-driving related tasks during automated driving in these kinds of vehicles, but that the automated system might ask drivers to resume manual driving in some situations (e.g., erased road markings). Participants were told that after reading each scenario, they would have to indicate how fast they would prefer the AC to drive, and how fast they would drive in the same situation. Finally, they were told that they would not receive any penalties for speeding (i.e., whether them or the AC was driving). The objective of this instruction was to prevent participants from limiting their speed due to the doubt of a possible sanction [4].

## Procedure

Participants clicked on the link they had received via social media or email, and carried out the study online. The experiment began with a general description of the study and a free and informed consent form. Instructions (' the "Scenario instructions" section above) were then given to participants. This was followed by a familiarization phase including a subset of 3 scenarios (see the "Scenario composition" section above), representing examples of favorable, unfavorable and mixed driving conditions. The subsequent experimental phase comprised the full 24 scenarios. Once this phase was completed, participants were asked to indicate their sex (*female, male, other, don't want to specify*), their past involvement in road accidents (*material damages only, with injured persons, with fatalities*), their car use frequency (< *once a week, 1–3 times a week, 3–5 times a week, > 5 times a week*), their past experience with automated driving systems (*none, automated cruise control, lane centering, automated cruise control coupled with lane centering*) and finally their level of trust in ACs on a scale ranging from 1 (*low*) to 5 (*high*).

## Hypotheses

According to the task-capability interface model of the driving process [14], speed adaptations allow drivers to reduce the difficulty of the driving task when facing high task demands (e.g., due to reduced visibility or proximity to other road users). Participants are thereby expected to report a slower manual driving speed in very rainy weather compared to clear weather

(Hypothesis (H) 1.a), and in high traffic congestion level compared to low level (H1.b). Participants are expected to follow the speed limits, and thus to report a slower manual driving speed in secondary roads compared to highway (H1.c), and slower in downtown compared to highway (H1.d) and to secondary road (H1.e).

As described earlier, previous studies have shown that most participants prefer automated driving-styles that match their own (e.g., [5]). Thus, since participants are expected to report a slower manual driving speed under these conditions (see hypotheses 1), they are also expected to prefer a slower AC speed in very rainy weather, high traffic congestion level, secondary roads (*vs*. highway) and downtown roads (*vs*. highway and secondary roads) (H2.a, H2.b, H2. c, H2.d and H2.e respectively).

Since participants are expected to prefer an automated driving-style that matches their own, the difference between speeds (i.e., reported manual driving speed and preferred ACs speed) is expected to be close to 0 (see the "Data analyses" and "Statistical power analyses" sections for more details) in each experimental condition:

- H3.a: highway, clear weather and low traffic congestion level.

- H3.b: highway, very rainy weather and low traffic congestion level.

- H3.c: highway, very rainy weather and high traffic congestion level.

- H3.d: highway, clear weather and high traffic congestion level.

- H3.e: secondary road, clear weather and low traffic congestion level.

- H3.f: secondary road, very rainy weather and low traffic congestion level.

- H3.g: secondary road, very rainy weather and high traffic congestion level.

- H3.h: secondary road, clear weather and high traffic congestion level.

- H3.i: downtown road, clear weather and low traffic congestion level.

- H3.j: downtown road, very rainy weather and low traffic congestion level.

- H3.k: downtown road, very rainy weather and high traffic congestion level.

- H3.l: downtown road, clear weather and high traffic congestion level.

However, participants with a low level of trust in ACs are expected to underutilize the automated driving system [11] and thus to prefer a lower ACs speed, leading to a higher negative difference with their reported manual driving speed. Conversely, participants with a high level of trust in ACs are expected to over-rely on the automated driving system [11] and thus to prefer a higher ACs speed, leading to a higher positive difference with their reported manual driving speed.

- H4: there is a positive linear relationship between trust in ACs and the difference between speeds.

## Data analyses

The data for hypotheses 1 and 2 was submitted to an analysis of variance (ANOVA). The main effects of the 3 variables (i.e., type of road, weather conditions and traffic congestion level) were examined, and their possible interaction effects on the participants' reported manual driving speed and preferred AC speed were explored. Since these interaction effects analyses were exploratory, only descriptive statistics were reported [25]. For hypotheses 3, a Two One-

Sided Tests (TOST) procedure was used to test for equivalence between the two speeds. The goal of this procedure is to reject the presence of a Smallest Effect Size of Interest (SESOI) [26], characterized here by a raw difference in km/h. Finally, for hypothesis 4, the data was submitted to a linear regression analysis in order to test the relationship between trust in ACs and the difference between speeds. All statistical analyses were carried out using R [27] and IBM SPSS (version 25.0) [28]. All reported statistics were cross-checked for consistency with statcheck.io [29].

## Statistical power analyses

The number of participants included in this study was based on time constraints. Sensitivity power analyses were run with G*Power (Version 3.1.9.6) [30,31] in order to determine the minimum reliably detectable effect sizes (i.e., $\beta = .20$; $\alpha = .05$ in this study). These analyses were performed at two different stages. First, prior to the study, to get a rough idea of the effect sizes that could be reliably detected, and to determine whether these effect sizes would be informative. Second, in order to have a more precise estimation, these analyses were performed based on the actual data collected. The results of these second analyses can be found in Table 1 for hypotheses 1 and 2, and show that the range of reliably detectable effects extends to small effects.

Regarding equivalence tests for hypotheses 3, a raw difference of 2.97 km/h was set as the SESOI in order to reach a minimum of 80% statistical power, and was calculated using the TOSTER R package [32]. It was based on the experimental condition where the standard deviation of the difference between speeds was the highest (i.e., SDIF = 10.31 in H3.a). As a result, the tests ran in the other experimental conditions used the same SESOI but with a higher statistical power (see Table 2).

Finally, a sensitivity analysis was also performed to determine the minimum reliably detectable effect size for hypothesis 4, indicating a slope of 2.02. Calculations were based on the standard deviation of trust in ACs, and on the standard deviation of the difference between speeds observed in the sample (i.e., respectively 1.14 and 9.61).

## Results

### Influence of the type of road, weather conditions and traffic congestion on reported manual driving speed

In line with hypothesis H1.a, the analysis revealed that the participants' reported manual driving speed was lower in very rainy conditions than in clear weather conditions, $F(1, 102) = 658.81$, $p < .001$, $\eta^2_p = .87$. In line with hypothesis H1.b, the analysis revealed that the

**Table 1. Minimum reliably detectable effect sizes for hypotheses 1 and 2.**

| Hypothesis | $\eta^2_p$ |
|---|---|
| 1.a | .01 |
| 1.b | .01 |
| 1.c/d/e | .02 |
| 2.a | .01 |
| 2.b | .001 |
| 2.c/d/e | .01 |

*Note.* Variations are due to the difference in the number of measurements and the correlations among repeated measures.

**Table 2. Statistical power for hypotheses 3' equivalence tests based on a raw difference of 2.97.**

| Hypothesis | SDIF | Statistical power |
|---|---|---|
| 3.a | 10.31 | 80% |
| 3.b | 9.49 | 87.48% |
| 3.c | 8.24 | 95.62% |
| 3.d | 9.61 | 86.52% |
| 3.e | 6.72 | 99.55% |
| 3.f | 7.66 | 97.81% |
| 3.g | 5.75 | 99.97% |
| 3.h | 6.10 | 99.9% |
| 3.i | 4.87 | 100% |
| 3.j | 5.84 | 99.96% |
| 3.k | 6.00 | 99.93% |
| 3.l | 5.46 | 99.99% |

*Note.* SDIF = Standard deviation of the difference.

participants' reported manual driving speed was lower in high congestion levels than in low congestion levels, $F(1, 102) = 184.97$, $p < .001$, $\eta^2_p = .64$. Finally, in line with hypotheses H1.c/d/e, the analysis revealed that the participants' reported manual driving speed was different between type of road conditions, $F(2, 101) = 7552.34$, $p < .001$, $\eta^2_p = .99$. It was lower ($p < .001$) in secondary road compared to highway (H1.c). It was lower ($p < .001$) in downtown compared to highway (H1.d). Finally, it was lower ($p < .001$) in downtown compared to secondary road (H1.e). For each hypothesis, see Table 3 for means, standard deviations and 95% confidence intervals.

The exploratory analyses revealed that the negative influence of adverse weather conditions on the participants' reported manual driving speed was different between type of road conditions (see Table 4). It was stronger for highway than for secondary roads and downtown. It was also stronger for secondary roads than for downtown.

The exploratory analyses revealed that the negative influence of higher levels of traffic congestion on the participants' reported manual driving speed was different between type of road conditions (see Table 5). It was stronger for highway than for secondary roads and downtown. It was also stronger for secondary roads than for downtown.

**Table 3. Participants' reported manual driving speed (in km/h) in the different experimental conditions.**

| Experimental condition | Mean | SD | 95% CI |
|---|---|---|---|
| **Weather conditions** | | | |
| Clear | 86.62 | 4.85 | [85.67, 87.57] |
| Very rainy | 75.41 | 5.77 | [74.28, 76.54] |
| **Traffic congestion level** | | | |
| Low | 83.40 | 4.63 | [82.50, 84.31] |
| High | 78.63 | 5.64 | [77.53, 79.73] |
| **Type of road** | | | |
| Highway | 117.84 | 7.01 | [116.47, 119.20] |
| Secondary | 79.00 | 5.47 | [77.94, 80.07] |
| Downtown | 43.21 | 5.14 | [42.05, 44.37] |

*Note.* SD = standard deviation; CI = confidence interval.

**Table 4. Participants' reported manual driving speed (in km/h) according to type of road × weather conditions.**

| Type of road | Weather conditions | Mean | SE | 95% CI |
|---|---|---|---|---|
| Highway | Clear | 128.45 | .73 | [127.00, 129.90] |
| | Very rainy | 107.22 | .79 | [105.65, 108.80] |
| Secondary | Clear | 82.69 | .58 | [81.55, 83.84] |
| | Very rainy | 75.32 | .67 | [73.99, 76.64] |
| Downtown | Clear | 48.72 | .49 | [47.76, 49.69] |
| | Very rainy | 43.69 | .61 | [42.48, 44.90] |

*Note.* SE = standard error; CI = confidence interval.

### Influence of the type of road, weather conditions and traffic congestion on preferred AC speed

In line with hypothesis H2.a, the analysis revealed that the participants' preferred AC speed was lower in very rainy conditions than in clear weather conditions, $F(1, 102) = 700.1$, $p < .001$, $\eta^2_p = .87$. In line with hypothesis H2.b, the analysis revealed that the participants' preferred AC speed was lower in high congestion levels than in low congestion levels $F(1, 102) = 206.06$, $p < .001$, $\eta^2_p = .67$. Finally, in line with hypotheses H2.c/d/e, the analysis revealed that the participants' preferred AC speed was different between type of road conditions, $F(2, 101) = 5744.49$, $p < .001$, $\eta^2_p = .98$. It was lower ($p < .001$) in secondary road compared to highway (H2.c). It was lower ($p < .001$) in downtown compared to highway (H2.d). Finally, it was lower ($p < .001$) in downtown compared to secondary road (H2.e). For each hypothesis, see Table 6 for means, standard deviations and 95% confidence intervals.

The exploratory analyses revealed that the negative influence of adverse weather conditions on the participants' preferred AC speed was different between type of road conditions (see Table 7). It was stronger for highway than for secondary roads and downtown. It was also stronger for secondary roads than for downtown.

The exploratory analyses revealed that the negative influence of higher levels of traffic congestion on the participants' preferred AC speed was different between type of road conditions (see Table 8). It was stronger for highway than for secondary roads and downtown. It was also stronger for secondary roads than for downtown.

### Equivalence tests: Speed personalization desirability for automated driving according to the type of road, weather conditions and traffic congestion

The same pattern of result has been observed for all experimental conditions, and is described in the following paragraph. For test statistics regarding individual hypothesis and their

**Table 5. Participants' reported manual driving speed (in km/h) according to type of road × traffic congestion level.**

| Type of road | Traffic congestion level | Mean | SE | 95% CI |
|---|---|---|---|---|
| Highway | Low | 120.95 | .74 | [119.49, 122.41] |
| | High | 114.72 | .79 | [113.16, 116.29] |
| Secondary | Low | 81.21 | .53 | [80.16, 82.25] |
| | High | 76.80 | .62 | [75.56, 78.04] |
| Downtown | Low | 48.05 | .49 | [47.08, 49.03] |
| | High | 44.36 | .60 | [43.17, 45.55] |

*Note.* SE = standard error; CI = confidence interval.

Table 6. Participants' preferred AV speed (in km/h) in the different experimental conditions.

| Experimental condition | Mean | SD | 95% CI |
|---|---|---|---|
| **Weather conditions** | | | |
| Clear | 86.62 | 4.85 | [85.67, 87.57] |
| Very rainy | 72.20 | 7.34 | [70.77, 73.64] |
| **Traffic congestion level** | | | |
| Low | 80.50 | 7.11 | [79.11, 81.89] |
| High | 75.38 | 7.65 | [73.88, 76.87] |
| **Type of road** | | | |
| Highway | 114.38 | 10.51 | [112.33, 116.44] |
| Secondary | 76.22 | 7.32 | [74.79, 77.65] |
| Downtown | 43.21 | 5.94 | [42.05, 44.37] |

*Note*. SD = standard deviation; CI = confidence interval.

Table 7. Participants' reported manual driving speed (in km/h) according to type of road × weather conditions.

| Type of road | Weather conditions | Mean | SE | 95% CI |
|---|---|---|---|---|
| Highway | Clear | 125.22 | 1.17 | [122.90, 127.54] |
| | Very rainy | 103.54 | 1.01 | [101.54, 105.55] |
| Secondary | Clear | 80.09 | .77 | [78.56, 81.62] |
| | Very rainy | 72.35 | .79 | [70.78, 73.92] |
| Downtown | Clear | 45.71 | .59 | [44.55, 46.88] |
| | Very rainy | 40.71 | .66 | [39.40, 42.02] |

*Note*. SE = standard error; CI = confidence interval.

Table 8. Participants' reported manual driving speed (in km/h) according to type of road × traffic congestion level.

| Type of road | Traffic congestion level | Mean | SE | 95% CI |
|---|---|---|---|---|
| Highway | Low | 118.01 | 1.10 | [115.82, 120.19] |
| | High | 110.75 | 1.08 | [108.62, 112.89] |
| Secondary | Low | 78.45 | .77 | [76.92, 79.98] |
| | High | 73.99 | .75 | [72.51, 75.47] |
| Downtown | Low | 45.03 | .55 | [43.95, 46.12] |
| | High | 44.36 | .60 | [43.17, 45.55] |

*Note*. SE = standard error; CI = confidence interval.

combination, see Table 9. For a visual representation of the combined effect, see Fig 1. The TOST procedure consisted of two one-sided tests for each hypothesis, and yielded nonsignificant results for the tests against the lower equivalence bound, $\Delta_L$, and significant results for the tests against the upper equivalence bound, $\Delta_U$. Although the tests against $\Delta_U$ indicate that one can reject differences at least as large as 2.97, the tests against $\Delta_L$ show that one cannot reject effects at least as extreme as -2.97. The equivalence tests are therefore non-significant. However, the 90% confidence intervals around the mean difference do not include 0, and thus the two-sided null hypothesis significance tests can be rejected. In summary, in each experimental condition, the participants' reported manual driving speed and the participants' preferred AC speed cannot be concluded as statistically equivalent– contrary to hypotheses 3 –but as significantly lower than 0.

**Table 9. Tests statistics for each hypothesis 3 and for their combination.**

| Hypothesis | TOST | NHST |
|---|---|---|
| H3.a | $t(102) = 0.34$, $p = .369$ | $t(102) = -2.59$, $p = .011$ |
| H3.b | $t(102) = -0.29$, $p = .613$ | $t(102) = -3.47$, $p < .001$ |
| H3.c | $t(102) = -1.41$, $p = .919$ | $t(102) = -5.07$, $p < .001$ |
| H3.d | $t(102) = -0.90$, $p = .815$ | $t(102) = -4.04$, $p < .001$ |
| H3.e | $t(102) = 1.22$, $p = .113$ | $t(102) = -3.27$, $p = .001$ |
| H3.f | $t(102) = -0.51$, $p = .695$ | $t(102) = -4.45$, $p < .001$ |
| H3.g | $t(102) = 0.71$, $p = .241$ | $t(102) = -4.54$, $p < .001$ |
| H3.h | $t(102) = -0.13$, $p = .550$ | $t(102) = -5.07$, $p < .001$ |
| H3.i | $t(102) = -0.38$, $p = .648$ | $t(102) = -6.57$, $p < .001$ |
| H3.j | $t(102) = 0.16$, $p = .439$ | $t(102) = -5.01$, $p < .001$ |
| H3.k | $t(102) = -0.18$, $p = .570$ | $t(102) = -5.20$, $p < .001$ |
| H3.l | $t(102) = 0.20$, $p = .420$ | $t(102) = -5.33$, $p < .001$ |
| Global | $t(102) = -0.19$, $p = .574$ | $t(102) = -5.42$, $p < .001$ |

*Note.* TOST = two one-sided tests; NHST = null hypothesis significance test; Global = test statistics combining all experimental conditions.

## Relationship between trust in automated cars and the difference between speeds

A simple linear regression was calculated to test if trust in ACs predicted the difference between speeds. The results of the regression indicated that the model explained 14.84% ($R^2_{adj}$ = .15) of the variance and that the model was significant ($F(1, 99) = 18.77$, $p < .001$). It was found that trust in ACs significantly predicted the difference between speeds (β = 2.01, 95% CI [1.09, 2.93], $p < .001$). The final predictive model was:

*Difference between speeds* $= -9.83 + (2.01 \times$ *trust in automated cars*$)$

*Note.* The difference between speeds is in km/h and trust in automated cars is on a scale ranging from 1 to 5.

## Discussion and conclusion

The overall objective of this study was to investigate the extent to which speed personalization of automated driving-styles is desirable with respect to various driving conditions and trust in a SAE level 3 automated car. In a first step, the influences of the type of road, weather

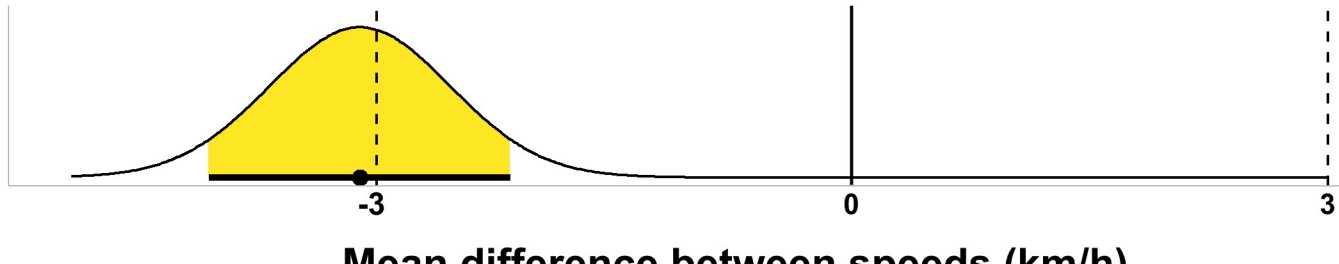

**Fig 1. Consonance density plot of the difference between the two speeds, combining all experimental conditions.** Note. The yellow area indicates the 90% confidence intervals from the two one-sided tests procedure. The dashed vertical black lines indicate the equivalence bounds (in raw scores) and the vertical black line indicates the nil null hypothesis.

conditions and traffic congestion were tested on the participants' reported manual driving speed and preferred AC speed. The results show an almost identical pattern of effects of the different variables of interest, whether it is on the participants' reported manual driving speed or on their preferred speed for the AC. Confirming H1.a/b and H2.a/b, both speeds were lower in adverse driving conditions (i.e., very rainy weather and high traffic congestion levels) compared to favorable ones. Additionally, confirming H1.c/d/e and H2.c/d/e, both speeds were lower in secondary road compared to highway, in downtown compared to highway and in downtown compared to secondary road. Overall, these results tend to be in line with the multiple comfort zones model [4] and the TCI model of the driving process [14]. They suggest that participants would prefer to reduce their speed in the most demanding driving conditions to reduce the difficulty of the driving task [14]. They also suggest that participants prefer the AC to behave like them under the different conditions, i.e., by reducing its speed in demanding situations and in relation to the type of road, and thus staying within the participants' comfort zones [4]. These results are in line with previous studies that showed that in most situations, personalized automated driving-styles are preferred [1,5,6], more accepted [6,7], and are perceived as more trustful [6,8], comfortable [6,8] and safe [6]. They also are in line with previous research, that showed that participants felt more comfortable in unfavorable driving conditions when the speed of the AC was lower than the speed limit [10]. Exploratory analyses revealed that the negative influence of adverse weather conditions and high levels of traffic congestion on both types of speed was stronger for highway than for secondary roads and downtown, and that it was stronger for secondary roads than for downtown. These results could be explained by the ascending order of speed limits, which are determinants of the driving task difficulty [14]. Indeed, the influence of speed on the demand of the driving task is low in downtown, a little higher in secondary roads and even higher on the highway. This could mean that in order to cope with the influence of weather conditions and the level of traffic congestion on the demand of the driving task, it is necessary to reduce the speed more substantially on the highway compared to downtown and secondary roads, and on secondary roads compared to downtown so that the difficulty of the driving task remains balanced.

The degree of proximity between the participants' reported manual driving speed and their preferred AC speed was tested in a second step. Contrary to expectations, the results showed that statistical equivalence between the two speeds could not be concluded in any of the experimental conditions, which goes against hypotheses 3. However, the results have also shown that the difference between the two speeds was significantly lower than 0 in all conditions. These results suggest that there is a tendency in participants to prefer the AC to drive slightly below their own driving speed.

The results observed for hypotheses 3 should be analyzed in light of those of H4. In fact, the relationship between trust in ACs and the difference between speeds was tested in this third step. The results showed that trust in ACs did indeed predict the difference between the two speeds, confirming H4. It should be noted that the size of the slope observed in this study was very slightly below the set minimum reliably detectable effect size. While the statistical hypothesis was confirmed, the initial prediction was that participants with a low level of trust in ACs would underutilize the automated driving system and participants with a high level would over-rely on the automated driving system [11]. The data support this first part, and indeed show that the lower the trust in ACs, the larger the negative difference between the participants' reported manual driving speed and their preferred speed for the AC. However, the data do not support the second part of the prediction, and show that for participants with the highest level of trust in ACs, the difference between the two speeds is actually around 0 (i.e., 0.22 km/h, following the predictive model described in the "Results" section), instead of being higher. Taken together, results for hypotheses 3 and H4 suggest that trust in ACs could play an

important role in the desirability of an automated driving system with personalized speed. Through the prism of the multiple comfort zones model [4], it can be hypothesized that comfort zones vary according to the degree of trust placed in the automated driving system. In order for the comfort zones concerning a drivenger's own driving and that of his AC to be similar, it could be necessary for the drivenger to have sufficient trust in the ability of the automated system to perform its driving task as well (e.g., comfortably) as he does. Future studies should replicate these findings with a higher sample size, allowing to test with sufficient power for equivalence in each experimental conditions, while controlling for trust in AC.

The results of the study need to be interpreted in consideration of a few methodological limitations. The main one is the fact that this study used a scenario-based experimental protocol. Indeed, although scenarios allow the participants to imagine specific situations, they do not allow to account for the great complexity that a road scene can represent. For example, real road traffic is composed of a multitude of vehicle types (e.g., cars, trucks, motorcycles, etc.) that could have different impacts on the participants' anticipated attitudes and behaviors. Hence, further studies should replicate these findings using simulator or real road experimental protocols. Another limitation of the present study that should be considered is the fact that trust was only considered in one dimension, that of *a priori* trust, and through a single item. Future studies should consider examining trust through multidimensional questionnaires (e.g., the Checklist for Trust between People and Automation [33]). Further studies should also consider investigating the impact of other variables inherent to the drivengers, such as manual driving-style [34] or driver locus of control [35].

Overall, the results of this study suggest that preferences for automated driving-style could vary as a function of drivengers-related variables (e.g., trust in ACs), but also as a function of environment-related variables (e.g., weather conditions). Additionally, the generalization to SAE level 3 of the benefits of automated driving-style personalization observed for SAE levels 4–5 may thus depend on the level of trust in ACs. This implies that from a user experience perspective, a "one for all" approach might not be desirable for the design of automated driving-styles. Conversely, a personalized approach seems preferable. This study also shows that asking participants to indicate their responses on two items rather than one for each scenario allows for a more detailed analysis of individuals' anticipated attitudes and behaviors. Indeed, in combination with the use of equivalence tests, this modification of the experimental protocol can provide evidence for the equivalence of two anticipated attitudes and/or behaviors in a given situation.

## Acknowledgments

We wish to thank all participants for supporting this research.

## Author Contributions

**Conceptualization:** Maxime Delmas, Valérie Camps, Céline Lemercier.

**Formal analysis:** Maxime Delmas.

**Funding acquisition:** Valérie Camps, Céline Lemercier.

**Investigation:** Maxime Delmas.

**Methodology:** Maxime Delmas, Valérie Camps, Céline Lemercier.

**Resources:** Maxime Delmas, Valérie Camps, Céline Lemercier.

**Supervision:** Valérie Camps, Céline Lemercier.

**Validation:** Maxime Delmas.

**Visualization:** Maxime Delmas.

**Writing – original draft:** Maxime Delmas.

**Writing – review & editing:** Valérie Camps, Céline Lemercier.

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
