## [Decision Letter · Decision Letter 0]

9 Jan 2023

PONE-D-22-33142Should my automated car drive as I do?: Investigating speed preferences of drivengers in various driving conditionsPLOS ONE

Dear Dr. Delmas,

Thank you for submitting your manuscript to PLOS ONE. After careful consideration, we feel that it has merit but does not fully meet PLOS ONE’s publication criteria as it currently stands. Therefore, we invite you to submit a revised version of the manuscript that addresses the points raised during the review process.

Your manuscript has received mixed reviews from four reviewers. Three of them had reservations about the methodology. In particular, the flow of the study and experiment scenarios could be clearly explained using more figures. Further, the significance of this work and the need for this research should be clarified. One reviewer questioned the sample size and sampling method. I hope that the authors could fix these issues and resubmit a revised version of this article.

We look forward to receiving your revised manuscript.

Kind regards,

Charitha Dias

Academic Editor

PLOS ONE

Journal Requirements:

2. Please change "female” or "male" to "woman” or "man" as appropriate, when used as a noun (see for instance https://apastyle.apa.org/style-grammar-guidelines/bias-free-language/gender).

Reviewers' comments:

Reviewer's Responses to Questions

**Comments to the Author**

1. Is the manuscript technically sound, and do the data support the conclusions?

Reviewer #1: Yes

Reviewer #2: Partly

Reviewer #3: Yes

Reviewer #4: Yes

2. Has the statistical analysis been performed appropriately and rigorously? 

Reviewer #1: Yes

Reviewer #2: No

Reviewer #3: Yes

Reviewer #4: Yes

3. Have the authors made all data underlying the findings in their manuscript fully available?

Reviewer #1: Yes

Reviewer #2: Yes

Reviewer #3: Yes

Reviewer #4: No

4. Is the manuscript presented in an intelligible fashion and written in standard English?

Reviewer #1: Yes

Reviewer #2: No

Reviewer #3: Yes

Reviewer #4: Yes

5. Review Comments to the Author

Reviewer #1: About the manuscript:

In the present study, the authors investigated the drivers' preferences for an SAE 3rd-level AV over different traffic conditions. After reviewing the manuscript, the following are my comments.

Comments:

In the introduction section, the authors listed the outcomes from the previous studies. But as a reader, it is essential to understand the research needs for carrying out the present work. I recommend that the authors dig into the research gaps and highlight the need for carrying out this work.

In the present case, authors used 24 scenarios for testing for an SAE 3rd-level AV. But in the current work, it needs to be clarified how those scenarios are selected and how they are adapted and validated is not clear. Further, how they imitate the SAE 3rd Level compared to other SAE level vehicles.

Further, I recommend that the authors present some pictorial representation of the scenarios. In the present form, the description is very brief, with limited details.

The title says, "Should my automated car drive as I do" but I didn't detail how the authors ascertained the human driver behaviour for comparing with automated driving.

I recommend that the authors present a research methodology section to explain the flow of the work. In the present form, it is tough to understand the flow of the work.

Reviewer #2: The current study aimed to investigate the benefits of automated driving styles personalization hold in various driving conditions.

The paper is not well written. As an example, the expectation is that the authors state the literature review and then state the objective and contribution of their work based on that. However, when they reach the objective expression, they have to write another part of literature review (i.e., lines 120-160). This was just an example; but here are more in the manuscript.

Sampling has a direct impact on the accuracy of interpretation of the results and its application. How did the authors of the article ensure that the number of 103 drivers with the stated characteristics can be representative of the community? Given how they were selected, how was it ensured that the selection was not biased?

As the authors of the article themselves have acknowledged, such studies have been based on either real road experiments or in a driving simulator. Using the scenario and participants' answers, although it may provide the possibility of asking various questions, but its validation is very crucial; especially when people's behavior is examined. How can the authors of this research ensure that the participants in reality show the same behavior as they said in response to the scenarios far from the conditions? I read the Procedure section but still doubt.

The article has several hypotheses and has been concluded with a very simple statistical analysis. I don't see a strong methodology or proper modeling in this paper. This is a major weakness for this research and publication in such a quality journal.

Reviewer #3: This study investigates the extent to which speed personalization of automated driving styles is desirable with respect to various driving conditions and trust in an SAE level 3 automated car. The research gap has been clearly defined. Further, authors have presented and discussed the experiment design and study outcomes thoroughly.

Reviewer #4: The authors present a clear and focused study that is well placed among the literature. To my knowledge (which I consider to be intermediate), the methods seem appropriate. The story builds logically to the conclusions. The authors state the limitations and suggest relevant future work.

The underlying data is probably available on the OSF website, but I don’t have access to confirm.

I suggest some minor edits in the attached file. A few comments ask for further explanation. Most comments are to help the authors to increase the readability. Note that some comments start with ‘Consider’. Their suggestions are optional, but I suggest them to improve readability according to my preferences. Authors, please use your judgement about the audience’s preferences.

Overall, the paper is good. I enjoyed reading and reviewing it. Well done!

6. PLOS authors have the option to publish the peer review history of their article (what does this mean?). If published, this will include your full peer review and any attached files.

Reviewer #1: **Yes: **narayana raju

Reviewer #2: No

Reviewer #3: No

Reviewer #4: **Yes: **Peter Stasinopoulos

---

## [Author Response · Author response to Decision Letter 0]

20 Jan 2023

As asked by the Academic Editor, the response to reviewers' comments/questions have been uploaded as a separate file named "Response to Reviewers".

---

## [Decision Letter · Decision Letter 1]

31 Jan 2023

Should my automated car drive as I do? Investigating speed preferences of drivengers in various driving conditions

PONE-D-22-33142R1

Dear Dr. Delmas,

We’re pleased to inform you that your manuscript has been judged scientifically suitable for publication and will be formally accepted for publication once it meets all outstanding technical requirements.

Kind regards,

Charitha Dias

Academic Editor

PLOS ONE

Additional Editor Comments (optional):

Reviewers' comments:

Reviewer's Responses to Questions

**Comments to the Author**

1. If the authors have adequately addressed your comments raised in a previous round of review and you feel that this manuscript is now acceptable for publication, you may indicate that here to bypass the “Comments to the Author” section, enter your conflict of interest statement in the “Confidential to Editor” section, and submit your "Accept" recommendation.

Reviewer #1: (No Response)

Reviewer #2: All comments have been addressed

2. Is the manuscript technically sound, and do the data support the conclusions?

Reviewer #1: (No Response)

Reviewer #2: Yes

3. Has the statistical analysis been performed appropriately and rigorously? 

Reviewer #1: (No Response)

Reviewer #2: Yes

4. Have the authors made all data underlying the findings in their manuscript fully available?

Reviewer #1: (No Response)

Reviewer #2: Yes

5. Is the manuscript presented in an intelligible fashion and written in standard English?

Reviewer #1: (No Response)

Reviewer #2: Yes

6. Review Comments to the Author

Reviewer #1: (No Response)

Reviewer #2: The authors could address may comments. So, I have no further comment. I suggest to accept the manuscript.

7. PLOS authors have the option to publish the peer review history of their article (what does this mean?). If published, this will include your full peer review and any attached files.

Reviewer #1: No

Reviewer #2: No

---

## [Editor Report · Acceptance letter]

1 Feb 2023

PONE-D-22-33142R1 

Should my automated car drive as I do? Investigating speed preferences of drivengers in various driving conditions 

Dear Dr. Delmas:

I'm pleased to inform you that your manuscript has been deemed suitable for publication in PLOS ONE. Congratulations! Your manuscript is now with our production department. 

Kind regards, 

on behalf of

Dr. Charitha Dias 

Academic Editor

PLOS ONE